# Can RUDAS Be an Alternate Test for Detecting Mild Cognitive Impairment in Older Adults, Thailand?

**DOI:** 10.3390/geriatrics6040117

**Published:** 2021-12-16

**Authors:** Manchumad Manjavong, Panita Limpawattana, Kittisak Sawanyawisuth

**Affiliations:** 1Department of Internal Medicine, Division of Geriatric Medicine, Faculty of Medicine, Khon Kaen University, Khon Kaen 40002, Thailand; manchu@kku.ac.th; 2Department of Internal Medicine, Division of Ambulatory Medicine, Faculty of Medicine, Khon Kaen University, Khon Kaen 40002, Thailand; kittisak@kku.ac.th

**Keywords:** cognitive impairment, cognitive screening test, minor neurocognitive disorders, Montreal Cognitive Assessment, neuropsychological tests

## Abstract

The Montreal Cognitive Assessment (MoCA) is the commonly used cognitive test for detecting mild cognitive impairment (MCI) in Thailand. Nevertheless, cultural biases and educational levels influence its performance. The Rowland Universal Dementia Assessment Scale (RUDAS) seems to lower the limitation of the MoCA. This study aimed to compare the performance of the RUDAS and the MoCA for the diagnosis of MCI and demonstrate the correlation between them. A cross-sectional study of 150 older participants from the outpatient setting of the Internal Medicine Department, Srinagarind Hospital, Thailand was recruited during January 2020 and March 2021. The diagnostic properties in detecting MCI of the RUDAS and the MoCA were compared. MCI was diagnosed in 42 cases (28%). The AUC for both RUDAS (0.82, 95% CI 0.75–0.89) and MoCA (0.80, 95% CI 0.72–0.88) were similar. A score of 25/30 provided the best cut-off point for the RUDAS (sensitivity 76.2%, specificity 75%) and a score of 19/30 for the MoCA had sensitivity and specificity of 76.2% and 71.3%. The Spearman’s correlation coefficient between both tests was 0.6. In conclusion, the RUDAS-Thai could be an option for MCI screening. It was correlated moderately to the MoCA.

## 1. Introduction

The global population is aging, leading to a higher incidence and prevalence rate of dementia [1]. This condition can cause numerous unfavorable outcomes, including physical, psychological, and economic consequences on the patients, their families, and society [2]. A bedside cognitive test is an important tool for the identification of cognitive impairment in medical practice since the diagnosis of this condition mainly depends on clinical judgement [3]. Mild cognitive impairment (MCI) is in the middle of normal cognition and dementia. Various causes of MCI including systemic conditions, neurological disorders, medications, and psychiatric conditions have been described. Its prevalence in older adults is approximately 6.7–25.2%, which is higher in increasing age and low education, and has more effects in men than women [1,4,5,6]. According to its pathogenesis, the outcomes can be generally categorized into three groups: return to normal aging, steadiness, or progression to dementia [7,8,9,10]. It progresses to dementia at a rate of about 5–17% annually [4,7,10,11]. Early detection of MCI is essential, as once it progresses to dementia it is usually incurable. Only two strategies to deal with dementia are available presently, which are symptomatic treatment and behavioral intervention [12]. Therefore, the detection of MCI with the use of possible interventions in the prevention or delaying the occurrence of dementia may diminish negative outcomes of dementia [13].

The most recommended bedside cognitive screening test for MCI is the Montreal Cognitive Assessment (MoCA) [14]. One systemic review reported its sensitivity of 80.48% and specificity of 81.19% at a cut-off point of 24/30 with the area under the Receiver Operating Characteristic (ROC) curve (AUC) of 0.846 (95% CI 0.823–0.868) [15], whereas its sensitivity and specificity were 80–100% and 50–76% at the cut-off point of 25/30 [6]. However, educational levels, different lifestyles, and ethnicities influence the performance of the MoCA [16,17]. For example, a report in Canada revealed that after correcting the MoCA total score for educational level, it lessened the sensitivity from 80% to 69% and slightly raised its specificity from 89% to 92%. At the optimal cut-off point of 24/30, it gave the sensitivity of 61% and the specificity of 97% [16]. The Cantonese Chinese version of the MoCA provided a sensitivity of 78% and specificity of 73% at a cut-off point of 22/30 [18]. The main limitation of MoCA is time-consuming than other cognitive screening tests, for example, the Rowland Universal Dementia Assessment Scale (RUDAS) or the Mini-mental state examination (MMSE [19,20]. Other bedside cognitive screening tools with lower limitations of the MoCA might be an option for use in bedside and outpatient settings.

The RUDAS is another cognitive screening test that was originated in multiethnic populations in Australia with good validity and reliability (both test–retest and inter-rater), [20,21,22,23,24]. Several versions in diverse languages exist, and it appears to not be influenced by sex or language; however, some studies from non-speaking English backgrounds showed that education could affect test performance [22,23]. In comparison with the MoCA, the RUDAS score had fewer variations than the MoCA with educational attainment (*p* < 0.01) [19]. There are limited studies about the diagnostic properties of the RUDAS in discriminating MCI from normal cognition [18,20,21,22,23,24]. A study in the Netherlands reported that the overall AUC of the RUDAS for differentiating normal cognition from MCI and dementia was 0.81, whereas the overall AUC of the MMSE was 0.77. Given the highest Youden index to determine the best cut-off point, a score at 21/30 had a sensitivity of 74% and specificity of 74% [20]. Furthermore, the effect of illiteracy on the RUDAS was smaller than on the MMSE [20]. One direct assessment study between the MoCA and the RUDAS in detecting MCI that involved patients from outpatient memory setting of London, Ontario, Canada, showed that at the cut-off point of 25/30, the MoCA showed a sensitivity of 95% and a specificity of 69%, whereas the RUDAS was 81% and 88%, respectively [19]. One longitudinal study in a multicultural rehabilitation setting in Australia which studied the association of the MoCA and the RUDAS with discharge outcomes showed that the RUDAS at <23/30 and the MoCA at <18/30 were related to nursing home placement (sensitivity of 52% and 57%, and specificity of 70% and 69%, respectively). There was a moderate correlation between those two tests (r = 0.571), and the RUDAS appeared to have a briefer administration time in both English- and non-English-speaking backgrounds [25].

The MoCA is the only screening test in differentiating cognitively intact versus patients with MCI in Thailand that had been studied and is practical in an outpatient setting though there are some limitations as mentioned earlier [26]. It is worthy to demonstrate the diagnostic accuracy of the RUDAS in our setting since the Thai version of the RUDAS (RUDAS-Thai) had good performance in detecting dementia, but there is no study in MCI patients [22,27]. The RUDAS appears to have lesser limitations than the MoCA. Thus, this study primarily aimed to compare the diagnostic property of the RUDAS-Thai and the MoCA-Thai in differentiating between old persons with intact cognitive function and MCI. The secondary aims were to exhibit the correlation between both tests and determine their optimal cut-off points in detecting MCI.

## 2. Materials and Methods

### 2.1. Study Design

A diagnostic cross-sectional study.

### 2.2. Subjects and Study Setting

The participants of this study were recruited from data from the “The performance of the Rowland Universal Dementia Assessment Scale (RUDAS), Recall test, and Mini-Cog in the screening of mild cognitive impairment” project [28,29], which was conducted during January 2020 and March 2021 at an outpatient setting of the department of Internal Medicine of the Srinagarind University Hospital, Khon Kaen University, Thailand. Subjects were older patients aged ≥60 years by Thai definition without evident acute illness that could influence the performance of the Thai version of RUDAS (RUDAS-Thai) and the MoCA (MoCA-Thai), for example, delirium, infection, acute cerebrovascular disease, and acute heart attack. Exclusion criteria included: subjects with a history of psychiatric conditions, congenital or acquired mental illnesses; subjects who used antipsychotic medications for a long period; subjects with severe dysfunction of vision, hearing, or movement; subjects with depressed mood based on the Thai version of the Patient Health Questionaire-9 (PHQ-9) >9 [30]; subjects with impaired instrumental activities of daily living (iADLs) determined by the Chula ADL index <9 [31], and the subjects who were disinclined to complete the tests or the ones who could not speak Thai. Figure 1 shows the study flow of this study.

### 2.3. Study Definition

#### Mild Cognitive Impairment (MCI)

MCI is diagnosed according to the definition of the Diagnostic and Statistical Manual of Mental Disorders (DSM-5) criteria which used the term “mild neurocognitive disorder” (mild NCD); however, the term “MCI” is more familiar in a clinical setting. The criteria included: (1) Evidence of minor cognitive deterioration from a prior performance in at least one cognitive domain, (2) The cognitive deficits do not affect the activities of daily living, (3) The deficits of cognitive function are not the result of delirium, and (4) The deficits of cognitive function are not better explained by other psychological conditions [32].

### 2.4. Cognitive Assessment Tools

#### 2.4.1. Rowland Universal Dementia Assessment Scale (RUDAS)

The RUDAS consists of 6 categories of items and measures multiple cognitive domains which are memory, praxis, visuoconstruction, language, and visuospatial domains. Administration of the test is easy which can be performed by a doctor, psychologist, or other trained persons, and completion of the questionnaires requires 10–15 min [21,33]. The highest score is 30, with a better score representing a better cognitive function. For the RUDAS-Thai, its sensitivity and specificity for dementia screening at the cut-off point of 23/30 (≤6 years of education) was 71.4% and 76.9%, respectively, whereas at the cut-off point of 24/30 (>6 years of education) the sensitivity and specificity were 77% and 70%, respectively [22,27].

#### 2.4.2. Montreal Cognitive Assessment (MoCA)

The MoCA was developed as a screening tool, particularly for detecting MCI. It takes about 10 min to administer. It measures several cognitive domains including visuospatial, executive function, naming, memory, attention, language, abstraction, delayed recall, and orientation. The total score is 30, with a greater score indicating greater cognitive performance. For the MoCA-Thai in detecting MCI, at the cut-off point of 25/30, the sensitivity was 80% and specificity was 80% [26].

### 2.5. Procedure

Baseline information of the subjects using convenience sampling was collected after obtaining written informed consent including age, gender, educational level, marital status, and underlying diseases. Then, the trained clinical investigator did the RUDAS-Thai and the MoCA-Thai in a random order where test–retest reliability of both tools was evaluated before administrating the tests to the subjects by rating scores to the same subject from the VDO recorder a week separately. Afterward, a geriatrician assessed MCI, according to DSM-5 criteria in the same period. Both the trained clinical investigator and the geriatrician had blinded the results of each other.

### 2.6. Sample Size Calculation

The area under the ROC curve was used to calculate the sample size based on the Hanley and McNeil’s method [34]. This technique adjusts the sample size till a satisfactorily small standard error of the AUC is reached. Ultimately, a sample size of 150 patients was acceptable and practical to conduct in a clinical setting at an AUC of 0.9 and standard error of 0.04.

### 2.7. Statistical Analysis

The statistical analyses were carried out with STATA software, version 10.0 (StataCorp, College Station, TX, USA). The Bland and Altman method was used to assess the intra-rater reliability of trained clinical investigator who did the RUDAS-Thai and MoCA-Thai. This method plots between the difference and means of the tests’ scores in diverse periods [35]. The implementation of this method was modified until the mean difference from the trained clinical investigator on two separate periods was not more than a score of 2 before collecting data in the main study.

Baseline characteristic data were presented as medians and inter-quartile ranges. Correlation between the RUDAS-Thai and MoCA-Thai scores was illustrated using a scatter plot with linear prediction and Spearman’s correlation. The overall performance of the RUDAS-Thai and the MoCA-Thai was summarized using the ROC curve. The AUC was estimated for each tool individually, along with their 95% confidence interval (CI). The diagnostic property and identification of the best cut-off points for MCI screening of the RUDAS-Thai and the MoCA-Thai were summarized by the sensitivity, specificity, predictive values, likelihood ratios, and AUC. The Youden index was used to determine the best cut-off point.

## 3. Results

### 3.1. Characteristics of Subjects

One hundred and fifty subjects were enrolled in this study where 28% (42/150 cases) had MCI. Table 1 shows baseline characteristics, MoCA, and RUDAS scores of the subjects. Women were predominated in the MCI group. Subjects with MCI were also older than the normal cognitive group, and diabetes mellitus was more prevalent in the MCI group. The MoCA and RUDAS scores were also lower in this group.

### 3.2. Correlation between Thai Version of RUDAS (RUDAS-Thai) and MoCA (MoCA-Thai)

The RUDAS-Thai scores had moderate correlation to the MoCA-Thai scores with a Spearman’s correlation coefficient of 0.6 (*p* < 0.00). Figure 2 shows a scatter plot of the correlation between those tests.

### 3.3. Screening Accuracy and Determination of the Best Cut-Off Points for MCI Detection

The overall performances of the RUDAS-Thai and MoCA-Thai in screening MCI using the AUC of the ROC curve were indifferent (*p* = 0.67) where the AUC of the RUDAS-Thai was 0.82 (95% CI 0.75–0.89) and the MoCA-Thai was 0.80 (95% CI 0.72–0.88) as presented in Figure 3. The diagnostic properties of the RUDAS-Thai and the MoCA-Thai for MCI screening with their different cut-off points are summarized in Table 2 and Table 3, respectively. The optimal cut-off points of the RUDAS-Thai and MoCA-Thai were ≤25/30 and ≤19/30, respectively.

## 4. Discussion

This study directly compared the diagnostic properties of the RUDAS-Thai and the MoCA-Thai in screening MCI among older adults in an outpatient setting where most studies focused their performance on screening dementia [21,22,23,36,37]. The finding of this study supports that the RUDAS score was moderately correlated to the MoCA score [25]. The results show that the overall performance of both tests was indifferent according to the AUC of the ROC curve. In comparison to a study in outpatient memory clinics that recruited patients from London, Ontario, Canada, it did not show the AUC of ROC curve of both tests, but it reported the sensitivity and specificity of the MoCA at the cut-off point of 25/30 of 95% and 69%, respectively, and for the RUDAS at the cut-off point of 25/30, it provided a sensitivity of 81% and a specificity of 88% [19]. For our finding, at the same cut-off point of the RUDAS-Thai (25/30), it demonstrated slightly lower sensitivity and specificity (76.2% and 75%), whereas the diagnostic property of the MoCA-Thai at the same cut-off point (25/30) did not have a good performance in MCI screening. Note that different cut-off points over 21/30 were not different due to a lack of discriminative properties as shown in Table 3. These findings can be explained by the disparity in cultural context and language preference setting. Since educational level does affect the diagnostic property of the RUDAS-Thai and MoCA-Thai, it might be another influenced factor; however, that study did not report the baseline educational level of the studied population [19].

Generally, the RUDAS-Thai and the MoCA-Thai had good psychometric properties in discriminating patients with MCI from normal cognition. According to Table 2, the RUDAS-Thai score of ≤25 provided good MCI screening property based on its performance whereas the MoCA-Thai score of 19 served as the best cut-off point, as shown in Table 3. This cut-off point of the MoCA-Thai was different from the previous study in Thailand that recommended its cut-off point at 25/30 [26]. The possible explanation is the earlier study conducted in Bangkok (central part of Thailand) and subjects with MCI had average greater years of education (11.3 years) where this study recruited subjects in the northeastern part of Thailand and most of the subjects with MCI had ≤6 years of education. The effect of cultural context and education could influence the performance of the MoCA-Thai [26].

As the psychometric properties of the RUDAS-Thai and the MoCA-Thai give similar results for MCI screening in geriatric patients and given the limitations of the MoCA-Thai including inconsistency in optimal cut-off point among Thai, greater variation with educational attainment, and longer time to administer, the RUDAS-Thai could be an option MCI screening tool in Thai geriatrics, and the cut-off point of 25/30 is recommended for indicating of having MCI.

Some limitations were noted in this study. Firstly, the prevalence of MCI is higher than the general population since the study venue is a tertiary care hospital where subjects appeared to have greater complex conditions which increase the risk of MCI. The generalizability of the result in other settings might be a concern. Secondly, the order of testing was random. Subjects might be exhausted from doing the second test, which might influence the test performance. Thirdly, MCI diagnosis was primarily on the clinical decision without the available laboratory test. Finally, a misclassification bias might have occurred due to the limitation of the cross-section study which lacks long-term follow-up and brain pathology.

## 5. Conclusions

The overall diagnostic property of the RUDAS-Thai and the MoCA-Thai for MCI screening is similar. Given the shorter administration time, less effect by education, and variation of cut-off points, the RUDAS-Thai might be more appropriate to administer in Thai geriatrics in detecting MCI. A score of 25 or lower is recommended as the best cut-off point to indicate the development of MCI.

## Figures and Tables

**Figure 1 geriatrics-06-00117-f001:**
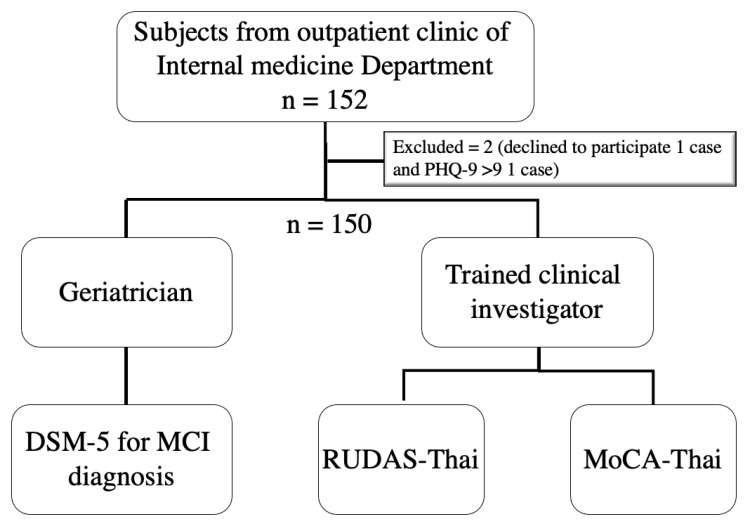
Flow of the study. Note: DSM-5; Diagnostic and Statistical Manual of Mental Disorders, MCI; mild cognitive impairment, RUDAS-Thai; Thai version of Rowland Universal Dementia Assessment Scale, MoCA-Thai; Thai version of Montreal Cognitive Assessment, PHQ-9; Thai version of the Patient Health Questionaire-9. Adapted from [29], Manjavong, M.; Limpawattana, P.; Sawanyawisuth, K. Performance of the Rowland Universal Dementia Assessment Scale in Screening Mild Cognitive Impairment at an Outpatient Setting. *Dement. Geriatr. Cogn. Disord Extra* **2021**, *11*, 181–188.

**Figure 2 geriatrics-06-00117-f002:**
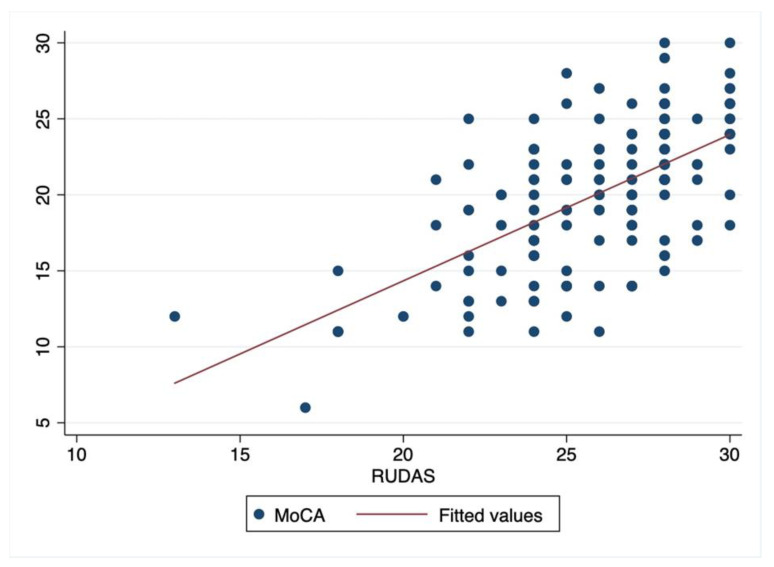
Scatter plot of correlation between RUDAS-Thai and MoCA-Thai.

**Figure 3 geriatrics-06-00117-f003:**
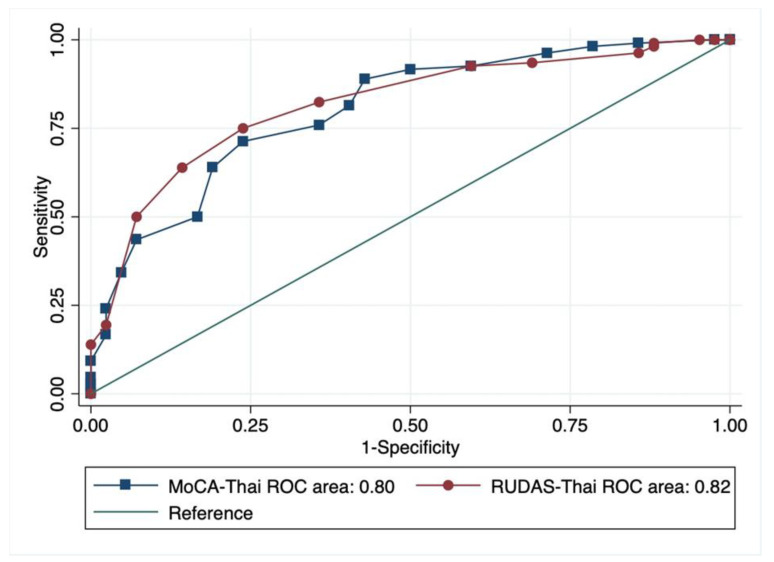
Comparison of the overall performance of the RUDAS-Thai and the MoCA-Thai using ROC curve.

**Table 1 geriatrics-06-00117-t001:** Baseline data of the patients.

Variables	Normal Cognition		MCI	
	*N* = 108 (72%)		*N* = 42 (28%)	
Age, med(IQR1,3)	67	(62.5,73.5)	72	(66,75)
Women, *n* (%)	53	(49.1)	25	(59.5%)
Years of education, *n* (%)				
- No	0	(0)	3	(7.1)
- ≤6 years	34	(31.5)	25	(59.5)
- 6 to 12 years	28	(25.9)	7	(16.7)
- >12 years	46	(42.6)	7	(16.7)
Marital status, *n* (%)				
- Single	3	(2.8)	2	(4.8)
- Married	82	(75.9)	26	(61.9)
- Divorce	8	(7.4)	1	(2.4)
- Widow	15	(13.9)	13	(30.9)
Underlying disease(s), *n* (%)				
- DM	43	(39.8)	23	(54.8)
- HT	81	(75)	33	(78.6)
- DLD	72	(66.6)	15	(35.7)
- CKD	27	(25)	4	(9.5)
- AF	7	(6.5)	4	(9.5)
- IHD	2	(1.9)	2	(4.8)
- CVA	5	(4.6)	4	(9.5)
- OSA	7	(6.5)	2	(4.8)
MoCA score, med (IQR1,3)	21.5	(19,24)	15.5	(13,19)
RUDAS score, med (IQR1,3)	27.5	(25.5,28)	24	(22,25)

Note: N; numbers of subjects, med; median, IQR; inter-quartile range, DM; diabetes mellitus, HTN; hypertension, DLD; dyslipidemia, CKD; chronic kidney disease, AF; atrial fibrillation, IHD; ischemic heart disease, CVA; cerebrovascular accident, OSA; obstructive sleep apnea, MoCA; Montreal Cognitive Assessment, RUDAS; Rowland Universal Dementia Assessment Scale. Adapted from [29], Manjavong, M.; Limpawattana, P.; Sawanyawisuth, K. Performance of the Rowland Universal Dementia Assessment Scale in Screening Mild Cognitive Impairment at an Outpatient Setting. *Dement. Geriatr. Cogn. Disord Extra* **2021**, *11*, 181–188.

**Table 2 geriatrics-06-00117-t002:** The RUDAS-Thai performance on screening for MCI at different cut-off points.

Cutoff Points	Sensitivity (%)	Specificity (%)	PPV (%)	NPV (%)	Youden Index	AUC	LR+	LR−
≤23	40.5	92.6	68	80	0.33	0.67	5.46	0.64
≤24	64.3	82.4	58.7	86.6	0.47	0.73	3.65	0.43
≤25	76.2	75	54.2	89	0.51	0.76	3.05	0.32
≤26	85.7	63.9	48	92	0.50	0.75	2.37	0.22
≤27	92.9	50	41.9	94.7	0.43	0.71	1.86	0.14

Note: AUC; area under the Receiver Operating Characteristic curve, CI; confidence interval, PPV; positive predictive value, NPV; negative predictive value, LR; likelihood ratio.

**Table 3 geriatrics-06-00117-t003:** The MoCA-Thai performance on screening for MCI at different cut-off points.

Cutoff Points	Sensitivity (%)	Specificity (%)	PPV (%)	NPV (%)	Youden Index	AUC	LR+	LR−
≤17	59.5	81.5	55.6	83.8	0.41	0.71	3.21	0.50
≤18	64.3	75.9	50.9	84.5	0.40	0.70	2.67	0.47
≤19	76.2	71.3	50.8	88.5	0.48	0.74	2.65	0.33
≤20	81	63.9	46.6	89.6	0.45	0.72	2.24	0.30
≤21	83.3	50	39.3	88.5	0.33	0.67	1.67	0.33

Note: AUC; area under the Receiver Operating Characteristic curve, CI; confidence interval, PPV; positive predictive value, NPV; negative predictive value, LR; likelihood ratio.

## Data Availability

No new data were created or analyzed in this study. Data sharing is not applicable to this article.

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
