# Peer review of "Can RUDAS Be an Alternate Test for Detecting Mild Cognitive Impairment in Older Adults, Thailand?"

_geriatrics, 2021, doi:10.3390/geriatrics6040117_

Round 1

Reviewer 1 Report

General comment:

This well written paper reports on the properties of the Rowland Universal Dementia Assessment Scale (RUDAS) in discriminating between patients with mild cognitive impairment (MCI) and healthy elderly persons and how it compares to the Montreal Cognitive Assessment (MoCA), an instrument widely recommended for the same purpose. The two instruments were found to have similar discriminating power. The findings are relevant to clinicians who have to deal with increasing numbers of elderly patients with cognitive complaints. A matter of concern is that most of the data related to the RUDAS have been published recently by the same group (Ref. #29). Permission by the publisher might be required for Table 1 and Figure 1, because they include the content of the corresponding table and figure of the former paper.

Specific (mostly minor) comments:

  1. Introduction, first paragraph, lines 35-36: The sentence beginning “Various causes of MCI …” seems to be incomplete.
  2. Introduction, first paragraph, lines 40-42: The sentence beginning “Early detection of cognitive impairment …” is not grammatically correct and therefore difficult to understand.
  3. Introduction, second paragraph, line 50: Please check, if reference #15 is the correct reference. The text is about the MoCA, but the reference is about MMSE.
  4. Introduction, third paragraph, lines 64 and 69: Reference #18 is not related to RUDAS.
  5. Introduction, third paragraph, line 76: “London, Ontario, and Canada” is not entirely correct. Ontario is a province of Canada, and London is a city in Ontario. The same is found in the Discussion, line 231.
  6. Materials and Methods, 2.3. Operational definition: The diagnosis of MCI was based on the DSM-5 definition. The correct DSM-5 term is mild neurocognitive disorder (mild NCD). The more familiar terms MCI and dementia are often used as synonyms to mild and major NCD, respectively. If this is done, the DSM-5 term should at least be mentioned in this paragraph.
  7. Materials and Methods, 2.4. Instrument, second paragraph (MoCA), line 147: “delayed decal” seems to be a misspelling. I suppose, this should read “delayed recall”.
  8. Materials and Methods, 2.7. Statistical analysis, last paragraph, last sentence, line 187: This sentence seems to be copied from the protocol. Here it should read “was” rather than “will be”.
  9. Results, 3.3. Screening accuracy …, line 211: Here it should read “Figure 3” rather than “Figure 2”. Likewise, in the caption to the figure showing the AUCs it should read “Figure 3” (line 215).
  10. Discussion, first paragraph, line 227: Reference #33 reports on data from a study in an inpatient population, not in an outpatient setting.
  11. Discussion, second paragraph, line 252: The second last sentence is misleading. It should read: “… the majority of the patients with MCI had 6 years or less education.” It was not the majority of the study sample.
  12. References, reference #26: The bibliographic information for reference #26 appears to be wrong. I could not locate it.

Author Response

Dear editor and reviewer 1,

Thank you for reviewing my manuscript. The revision has been done according to your recommendations in the revised manuscript and I have answered you point-by-point with the following:

General comment:

This well written paper reports on the properties of the Rowland Universal Dementia Assessment Scale (RUDAS) in discriminating between patients with mild cognitive impairment (MCI) and healthy elderly persons and how it compares to the Montreal Cognitive Assessment (MoCA), an instrument widely recommended for the same purpose. The two instruments were found to have similar discriminating power. The findings are relevant to clinicians who have to deal with increasing numbers of elderly patients with cognitive complaints. A matter of concern is that most of the data related to the RUDAS have been published recently by the same group (Ref. #29). Permission by the publisher might be required for Table 1 and Figure 1, because they include the content of the corresponding table and figure of the former paper.

Reply: Thank you for your comment. This manuscript was a sub-study under the project entitled of “The performance of the Rowland Universal Dementia Assessment Scale (RUDAS), Recall test and Mini-Cog in the screening of mild cognitive impairment” that has been described in the materials and methods part (2.2 Subjects and study setting). The participants; therefore, were the same group (as shown in Table 1) but there was more information about MoCA score. However, the objective of each project was different. Fig 1 in this manuscript compared the performance between the MoCA and the RUDAS but in the ref#29 did not. Therefore, I think that permission from the publisher of the ref#29 is not required.

Specific (mostly minor) comments:

  1. Introduction, first paragraph, lines 35-36: The sentence beginning “Various causes of MCI …” seems to be incomplete.

Reply: I have revised the sentence as “Various causes of MCI including systemic conditions, neurological disorders, medications, and psychiatric conditions have been described”.

  1. Introduction, first paragraph, lines 40-42: The sentence beginning “Early detection of cognitive impairment …” is not grammatically correct and therefore difficult to understand.

Reply: I have revised the sentence as “Early detection of MCI is essential as once it progresses to dementia, it is usually incurable.”

  1. Introduction, second paragraph, line 50: Please check, if reference #15 is the correct reference. The text is about the MoCA, but the reference is about MMSE.

Reply: I apologize for the mistake. The actual reference was “Ciesielska, N.; SokoÅ‚owski, R.; Mazur, E.; Podhorecka, M.; Polak-Szabela, A.; KÄ™dziora-Kornatowska, K. Is the Montreal Cognitive Assessment (MoCA) Test Better Suited than the Mini-Mental State Examination (MMSE) in Mild Cognitive Impairment (MCI) Detection among People Aged over 60? Meta-Analysis. Psychiatr. Pol. 2016, 50, 1039–1052.  I have edited it in the reference part.

  1. Introduction, third paragraph, lines 64 and 69: Reference #18 is not related to RUDAS.

Reply: I apologize for the mistake. The reference#18 was deleted from this paragraph.

  1. Introduction, third paragraph, line 76: “London, Ontario, and Canada” is not entirely correct. Ontario is a province of Canada, and London is a city in Ontario. The same is found in the Discussion, line 231.

Reply: Thank you for your comment. I have revised the sentence as “…London, Otario, Canada".

  1. Materials and Methods, 2.3. Operational definition: The diagnosis of MCI was based on the DSM-5 definition. The correct DSM-5 term is mild neurocognitive disorder (mild NCD). The more familiar terms MCI and dementia are often used as synonyms to mild and major NCD, respectively. If this is done, the DSM-5 term should at least be mentioned in this paragraph.

Reply: Thank you for you comment.  I have revised the sentence as “MCI is diagnosed according to the definition of the Diagnostic and Statistical Manual of Mental Disorders (DSM-5) criteria which used the term “mild neurocognitive disorder (mild NCD); however, the term “MCI” is more familiar in a clinical setting.”.

  1. Materials and Methods, 2.4. Instrument, second paragraph (MoCA), line 147: “delayed decal” seems to be a misspelling. I suppose, this should read “delayed recall”.

Reply: I have edited the typo as your comment.

  1. Materials and Methods, 2.7. Statistical analysis, last paragraph, last sentence, line 187: This sentence seems to be copied from the protocol. Here it should read “was” rather than “will be”.

Reply: I have changed it as your recommendation as “The statistical analyses were done with STATA software, version 10.0 (StataCorp, College Station, TX, USA)”.

  1. Results, 3.3. Screening accuracy …, line 211: Here it should read “Figure 3” rather than “Figure 2”. Likewise, in the caption to the figure showing the AUCs it should read “Figure 3” (line 215).

Reply: I have edited the mistake as your comment.

  1. Discussion, first paragraph, line 227: Reference #33 reports on data from a study in an inpatient population, not in an outpatient setting.

Reply: I have removed it as your suggestion.

  1. Discussion, second paragraph, line 252: The second last sentence is misleading. It should read: “… the majority of the patients with MCI had 6 years or less education.” It was not the majority of the study sample.

Reply: Thank you for your suggestion. I have revised the sentence as “The possible explanation is the earlier study conducted in Bangkok (central part of Thailand) and subjects with MCI had average greater years of education (11.3 years) where this study recruited subjects in the northeastern part of Thailand and most of the subjects with MCI had <6 years of education”.

  1. References, reference #26: The bibliographic information for reference #26 appears to be wrong. I could not locate it.

Reply: I apologize for the mistake. I have changed the rederence#26 to “Tangwongchai, S.; Phanasathit, M.; Charernboon, T.; Akkayagorn, L.; Hemrungrojn.; Nasreddine, Z.A. The Validity of Thai Version of The Montreal Cognitive Assessment (MoCA-T). Dement Neuropsychol. 2009, 3,172.” The full article can be found at https://www.scienceopen.com/document?vid=72c682ca-4282-40d6-8fa5-83872b3c9c86.

Reviewer 2 Report

This article is of great clinical use value. The RUDAS scale is interesting and a
promising option for those who have lower literacy or less time to answer
questionnaires. Overall the manuscript, from Introduction to the Conclusion, is easy to read, the text is well written, concise and logical and, conceptually and methodologically, relevant and correct.

I have only one point to make: why have the authors not also linked DSM-5 with RUDAS and MoCA?

Page 8, line 231, remove and from "London, Ontario, and Canada".

Author Response

Dear editor and reviewer 2,

Thank you for reviewing my manuscript. The revision has been done according to your recommendations in the revised manuscript and I have answered you point-by-point with the following:

This article is of great clinical use value. The RUDAS scale is interesting and a
promising option for those who have lower literacy or less time to answer
questionnaires. Overall the manuscript, from Introduction to the Conclusion, is easy to read, the text is well written, concise and logical and, conceptually and methodologically, relevant and correct.

I have only one point to make: why have the authors not also linked DSM-5 with RUDAS and MoCA?

Reply: Thank you for your comment. The diagnostic criteria for MCI in this study was based on the definition of the DSM-5 criteria which used the term “mild neurocognitive disorder (mild NCD); however, the term “MCI” is more familiar in clinical setting. Therefore, the performance of the RUDAS and MoCA was linked to the DSM-5 criteria. I have added more information in the materials and methods part (2.3 Study definition section) to make it clearer.

Page 8, line 231, remove and from "London, Ontario, and Canada".

Reply: I have edited as your recommendation.